# Development of Digital Fetal Heart Models with Virtual Ultrasound Function Based on Cardiovascular Casting and Computed Tomography Scan

**DOI:** 10.3390/bioengineering9100524

**Published:** 2022-10-05

**Authors:** Qi Yang, Jiancheng Han, Rui Wang, Tong Zhang, Yingying Zhang, Jingyi Wang, Lei Xu, Zengguang Hou, Shuangyi Wang, Yihua He

**Affiliations:** 1Echocardiography Medical Center, Beijing Anzhen Hospital, Capital Medical University, Beijing 100029, China; 2Maternal-Fetal Medicine Center in Fetal Heart Disease, Beijing Anzhen Hospital, Beijing 100029, China; 3Department of Network Intelligence, Peng Cheng Laboratory (PCL), Shenzhen 518055, China; 4State Key Laboratory of Management and Control of Complex Systems, Chinese Academy of Sciences, Institute of Automation, Beijing 100190, China

**Keywords:** congenital heart disease, fetal echocardiography, digital twin, cardiovascular casting, virtual ultrasound

## Abstract

Despite recent breakthroughs in diagnosis and treatment, congenital heart defects (CHDs) continue to be the leading cause of death among newborns. Fetal echocardiography is the most effective and non-invasive method for the prenatal diagnosis of CHDs. However, the challenge of obtaining standard views can lead to a low diagnostic accuracy. To explore new methods for training, the combined use of cardiovascular casting, computed tomography (CT) scanning, and virtual ultrasound generation methods was studied to preserve the cardiac structures of a fetus in digital form. The feasibility of the proposed workflow was verified by testing three fetal heart specimens collected after the termination of pregnancy. As a result, the anatomical structures were imaged clearly by a CT scan after cardiovascular casting, and the virtually generated ultrasound images based on the use of the Public software Library for UltraSound imaging research (PLUS) toolkit successfully demonstrated both the standard views and the views with diagnostic values for the visualization of the cardiovascular structures. This solution provides great data extensibility while being simple and cost-effective for end users. Therefore, the proposed method could provide a promising educational system for trainees to understand standard views of fetal echocardiography and the corresponding anatomical correlations.

## 1. Introduction

Congenital heart disease (CHD) includes abnormalities in the heart structure that occur before birth [1]. It is the most common congenital anomaly, accounting for approximately 0.8% of live births [2,3], and ranks first among birth defects [4]. Despite significant advances in diagnosis and treatment in recent years, CHD remains the most common cause of death in newborns. An accurate prenatal diagnosis of cardiac defects is crucial for perinatal management and treatment [5,6,7]. Fetal echocardiography is currently the most effective and non-invasive method for the prenatal diagnosis of CHD. However, the challenge of obtaining standard views with diagnostic values generally leads to a low accuracy. According to studies on the use of general ultrasonography [8,9], one of the most significant hurdles to a reliable image acquisition is the lack of training. As the views are generally more difficult to comprehend for fetal echocardiography due to the complicated relationship between the two-dimensional (2D) slices and the three-dimensional (3D) anatomical structures, the problem with training is exacerbated and requires more effort to resolve.

Unfortunately, current training methods for fetal echocardiography are relatively limited to the theoretical level and cannot fully meet the needs of skill operation training. For operation practice, the fetal ultrasound training phantom is an ideal choice that can provide a tissue-equivalent and anatomically appropriate model for the ultrasonic scanning of fetal cardiovascular structures and can enable training in standard view acquisition techniques. These systems, such as the commercial products from CIRS Inc. (Norfolk, VA, USA) and Kyoto Kagaku Co., Ltd. (Kyoto, Japan), normally involve great manufacturing expenses. Regular hospitals, training institutes, and medical schools are typically unable to afford these systems for individual training because of the high cost. Moreover, the physical form of a fetal phantom contains only one set of data, which is often the normal heart. Therefore, practicing for different cases is not possible.

To explore new methods for training and education, we consider postmortem imaging as an opportunity to preserve the heart structures in digital form and cardiovascular casting as an established method of anatomical preparation that is capable of exhibiting the 3D architecture of the cardiovascular system in a physical form. Since the 1990s, magnetic resonance imaging [10,11] and micro-computed tomography [12,13,14] have been used for postmortem cardiac imaging. Although these methods can display delicate intracardiac and extracardiac structures, the spatial configuration of the complex malformation of the cardiovascular system can be demonstrated more stereoscopically and vividly by the vascular corrosion technique [15,16,17,18,19]. Moreover, the cardiovascular systems are initially separated during the casting process, making the next step of precise segmentation easier. At Beijing Anzhen Hospital, Capital Medical University, cardiovascular casting can be regularly performed using fetal specimens, and many fetal heart models with different abnormalities have been preserved through years of accumulation. It is of interest to investigate whether we can digitize the cast models and use them for ultrasound educational purposes.

In this paper, we aim to investigate a novel digital twin approach using cardiovascular casts to preserve the original hearts, CT scans for digitization, and simulation algorithms to generate virtual ultrasound images. This method enables the digital reproduction of fetal echocardiography, allowing the user to operate the ultrasound probe in a virtual environment and to learn ultrasound standard views and anatomical relationships. Owing to its characteristics, this method has a good data extensibility and is convenient and affordable for the end user to access. Herein, we report the methods for the workflow and results of the test with three fetal heart specimens.

## 2. Materials and Methods

### 2.1. Cardiovascular Casting

The three fetuses were diagnosed with fetal cardiac function or structural abnormality by fetal echocardiography during the second trimester of the pregnant women, and the mothers finally chose to terminate the pregnancy and to induce labor in our hospital. After informed consent, the parents agreed to donate the fetal remains to the center for further research. Inclusion criteria: all fetuses whose bodies were donated. In this study, one normally structured fetal heart specimen with heart failure and two specimens prenatally diagnosed with CHDs were collected after the termination of pregnancy. The gestational ages were 26, 26, and 25 weeks. The diagnostic results before the pregnancy termination are presented in Table 1. The study was conducted with the formal consent of the parents and was approved by the Medical Ethics Committee of Beijing Anzhen Hospital, Capital Medical University (Approval No. 2019030). The abdomen and thorax were opened along the median line and costal margins with an inverted y-shaped incision and a midline abdominal incision. The heart and lungs were exposed after the removal of the thymus. A plastic catheter was inserted into the umbilical vein. The cardiovascular system of the specimen was then rinsed with normal saline via the umbilical vein to drain the residual blood, and the casting material was injected into the cardiovascular system via the umbilical vein at a steady pressure and speed. When the casting material solidified, the specimen was soaked in a potassium hydroxide solution to dissolve the surrounding tissues. Two days after corrosion, the cast specimen was cleaned and preserved for further study.

### 2.2. Creation of Digital Heart

To produce volumetric data, the cast models were scanned by CT, and the digital hearts were generated by manual annotation and segmentation according to the CT image data. The three casts were respectively placed flat in a SOMATOM Definition Flash scanner and scanned while setting the slice thickness at 0.6 mm and distance at 0.4 mm. The CT scan occurred one year after the establishment of the cast. Since the relevant structures have been preserved by the casts, the scan time does not affect the results. With the scanned images exported and stored in the Digital Imaging and Communication in Medicine format, the required structures of the hearts were semi-automatically segmented using active contour methods and were manually delineated by medical staff using ITK-SNAP (Ver. 3.8.0). The anatomy of the segmented heart extracted from the CT image volume included the left atrium (LA), left atrial appendage (LAA), right atrium (RA), right atrial appendage (RAA), left ventricle (LV), right ventricle (RV), aorta (AO, including ascending aorta, aortic arch (Arch) and descending aorta (DAO), aortic branches (AoBs), pulmonary artery (PA, including pulmonary trunk, left pulmonary artery (lPA) and right pulmonary artery (rPA), ductus arteriosus (DA), superior vena cava (SVC), and inferior vena cava (IVC). The reliability of the segmentation was validated by another cardiac imaging specialist. 

### 2.3. Generation of Virtual Ultrasound

The proposed workflow utilizes the Public software Library for UltraSound (PLUS) imaging research toolkit [20] to generate simulated ultrasound images and the open-source software 3D Slicer (Ver. 4.10.2 r28257) as the main workspace and graphical user interface. The surface mesh model of the segmented fetal heart was imported and exhibited on the main display along with a virtual ultrasound probe and the simulated ultrasound image generated based on the current transducer location. This was achieved using an ultrasound image simulation module implemented in PLUS [21]. A network communication module, known as OpenIGTLink [22], was employed for communication between the main workspace in 3D Slicer and the PLUS toolkit. The homogeneous linear transformation in matrix form, to represent the current transducer location, was transferred from the main workspace to the PLUS toolkit, and the generated ultrasound image data were then sent back from the PLUS toolkit to the main workspace. In the main workspace, the current transducer location can be adjusted by the user using the sliders, including changes in both the position and orientation of the virtual ultrasound probe.

With the imported surface mesh model of the heart and the transformation representing the spatial relationship in between, the heart model was transformed into image coordinates, and the intersecting region was calculated and displayed as a 2D binary image. During this process, the running script of the PLUS toolkit calculated the locations of the virtual scan lines in the image coordinate system based on the specified size of the image, which was predefined in a configuration XML file. The intersection locations of the cardiac structures and the produced scan lines were determined using a binary space partitioning tree. The scan line was divided into segments by the crossover locations, which were subsequently filled with a grey value. The material parameters defined in the configuration XML file were used to compute the intensity. The scan lines were then transformed into a conventional brightness mode (B-mode) ultrasound image.

### 2.4. Clinical Feasibility Test

We designed and conducted a user study to evaluate the feasibility of the digital model with the ultrasound generation function and its clinical acceptance. The procedure of the study is shown as follows: (1) Participants were informed of the purpose and the procedure and then instructed to use the virtual ultrasound model; (2) The operator was allowed to become familiar with using the software for a certain time (10 min), to designate a view, and to observe whether the subject could find the view accurately and quickly; (3) In the end, participants were asked to fill out a questionnaire, grading statements concerning the complexity, visuality, usability, applicability, and similarity using a 5-point Likert scale (1: strongly disagree; 3: neutral; 5: strongly agree).

In the post-experiment questionnaire, five statements were given for the participants to grade: (1) “It was easy for me to learn how to use the model and complete the task” (complexity); (2) “I felt that the images of virtual fetal hearts and virtual ultrasound views were clear and vivid” (visuality); (3) “It was easy for me to move the virtual probe to the guiding position and transform the specified angle quickly to find the specified ultrasound view” (usability); (4) “I felt this model is conducive to understanding the anatomical structure of the fetal heart and the relationship between 2D ultrasound views and the 3D geometry of the heart” (applicability); (5) “I felt that the virtual ultrasound images were very similar to real ultrasound images” (similarity).

## 3. Results

Three cast models were successfully obtained and vividly demonstrated the 3D anatomy of the cardiovascular system. The results of the casting with the major anatomical structures are shown in Figure 1.

To better reflect the cardiovascular structure, the three specimens were perfused with different doses of pigment for comparison, resulting in three different colored casts. As shown in Figure 1, Case 1 demonstrates the casting of the normal structure of the heart. In Case 2, the ascending AO is divided into two branches (the left and right aortic arches) that form a complete vascular ring. In Case 3, the RV is significantly smaller than the LV, and there is also a channel between the LV and RV. The AO is located anteriorly and arises from the RV. The PA is located posteriorly, and the initial part of the PA is not connected to the ventricle.

The corresponding anatomical structures were imaged clearly by CT scanning of the cardiovascular casts, and reconstructed 3D digital models were successfully established after segmentation. The digital models could distinctly display the anatomical and spatial structures of the chambers and major vessels. Using the virtual ultrasound generation method, three scenes were prepared in 3D Slicer, allowing for the visualization of the 3D structures of the heart, for the adjustment of the virtual ultrasound probe, and for obtaining the simulated 2D ultrasound views. The three reconstructed 3D digital models and the simulated ultrasound acquisition scenes are shown in Figure 2.

Using the final scenes, eight main standard views (four-chamber view, left ventricular outflow tract view, right ventricular outflow tract view, three-vessel view, three vessels and trachea, aortic arch view, ductal view, and bicaval view) [23,24] and additional views in which the lesion was adequately demonstrated could be obtained. Examples of standard views for the normal heart are shown in Figure 3, and example views with diagnostic values for the two abnormal hearts are shown in Figure 4. As shown in Figure 4a, there is an abnormal vessel between the DA and the aortic arch in the three vessels and the trachea view. In Figure 4b, the probe moves toward the cephalic side of the fetus from the three-vessel trachea view to show a cross-section of the double aortic arches. In Figure 4c, the double aortic arch connects the descending AO at the coronal level. In Figure 4d, the RV is significantly smaller than the LV in the four-chamber view. In Figure 4e, there is a channel between the left and right ventricles (indicated by the arrow), and the AO arises from the RV. In Figure 4f, the initial part of the PA is not connected to the ventricle (indicated by the arrow).

As for the clinical feasibility test, 16 volunteers specialized in ultrasound medicine (nine graduate students and seven resident doctors) participated. The results of this subjective evaluation are shown as a radar plot in Figure 5. On average, the users believed that the virtual ultrasound model was not complicated to use and had a good visibility and strong applicability, but it was not easy to adjust the position and angle of the virtual probe by using the mouse to drag the axes. Moreover, feedback from the users showed that in the virtual ultrasound views, the proportional relationship between the heart structures showed variations from the real ultrasound images.

## 4. Discussion

In this study, the digitization of fetal hearts was achieved with the combined use of cardiovascular casting, a CT scan, and virtual ultrasound generation methods. In the workflow, cardiovascular casting can preserve anatomical structures by simulating the filling of blood, thus laying the foundation for further digital processing. As can be visually observed from the results, the spatial relationship between the 3D geometries and the morphological structures of the heart and major vessels was clearly displayed using this technology. With a cast model, a CT scan can be performed at any moment in the future to obtain the 3D volumetric information needed to digitize the heart. As can be observed from the results, the virtual model can demonstrate the 3D anatomical structures of the heart and the 3D spatial morphology of the major vessels. As a digital form of the casts, the virtual model is also capable of intuitively showing the origin, course, and interrelations of major vessels. Therefore, normal and pathological anatomical information was displayed and retained in a digital form.

However, as the users reported, it should also be noted that the reconstructed digital heart is not entirely consistent with the real fetal heart. Due to the imbalance of perfusion pressure, the inner diameters of some chambers and vessels near the perfusion entrance were larger than the actual structures, while the inner diameters of some chambers and vessels far away from the perfusion inlet were smaller. Additionally, different cardiovascular structures have varying capacities to withstand pressure, which leads to an inconsistent expansion of the chambers and major vessels. Consequently, the proportional relationship between the cardiovascular system structures can change. This effect needs to be further investigated and calibrated by comparing the CT scans of the cast models with the original hearts. It is feasible for proportional relationships to be accordingly calibrated by processing and scaling different structures of the heart, although more detailed studies need to be performed separately.

With the generation of virtual ultrasound images, the eight main standard views and views through which the lesion is adequately illustrated can be successfully found by virtually controlling the ultrasound transducer in the software, making it a useful tool for hands-on operations and skill practice. In addition, the direct visualization of the 3D cardiovascular structures and the position of the virtual probe can help establish correlations between 3D spatial structures and 2D anatomical planes. The above results were also confirmed by 16 professionals in the questionnaire session, which demonstrates that the proposed method can provide a direct aid to clinical training and case analyses.

Based on the subjective evaluation results, the value of the method proposed in this study was recognized by the participants, but there is a need for further research to optimize the position and angle transformation of the virtual probe to make the operator’s experience more similar to the procedure of a real fetal echocardiography. Moreover, it should also be noted that not all real fetal ultrasound images are as clear as the simulated virtual images because of the maternal abdominal wall thickness, amniotic fluid volume, and fetal position and posture in utero. As a result, the virtual system proposed in this work is primarily intended for educational purposes, while actual hands-on training remains indispensable. Nevertheless, in our future work, we will improve our virtual system by adding maternal abdominal wall conditions and allowing for the movements of the heart.

## 5. Conclusions

This paper presents a novel approach to preserving the 3D anatomical structural features of the fetal heart and to further achieving digital visualization and virtual ultrasound imaging capabilities for ultrasound education and hands-on training purposes. The workflow based on cardiovascular casting and CT scanning has demonstrated the feasibility of generating a digital heart and virtual ultrasound, although the calibration of the proportional relationships between the heart structures is necessary. In conclusion, the proposed system provides a simulation environment for fetal echocardiography, allowing the user to learn standard views and anatomical correlations while operating the ultrasound probe virtually. This method offers a strong data extensibility and is convenient and economical for end users. Our future work will focus on improving user interactions and enhancing authenticity.

## Figures and Tables

**Figure 1 bioengineering-09-00524-f001:**
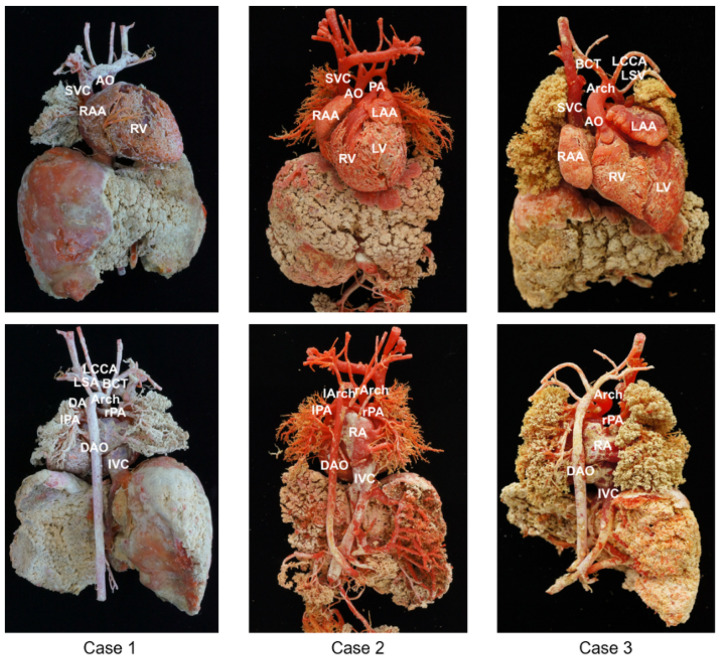
Results of the three cast models with the anterior and posterior views shown. AO—aorta; Arch—aortic arch; BCT—brachiocephalic trunk; DA—ductus arteriosus; DAO—descending aorta; IVC—inferior vena cava; LAA—left atrial appendage; LCCA—left common carotid; LV—left ventricle; LSA—left subclavian artery; PA—pulmonary artery; RA—right atrium; RAA—right atrial appendage; RV—right ventricle; SVC—superior vena cava.

**Figure 2 bioengineering-09-00524-f002:**
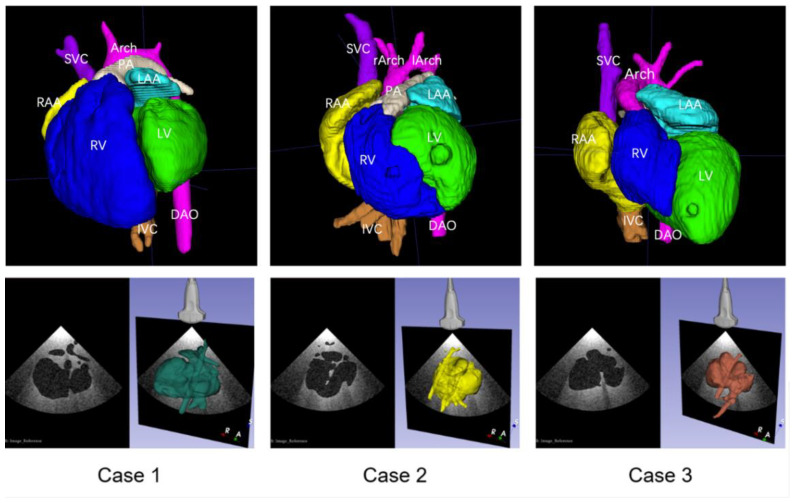
Reconstructed 3D digital models with major structures segmented after computed tomography scan, and simulated ultrasound acquisition scenes for the three cast models.

**Figure 3 bioengineering-09-00524-f003:**
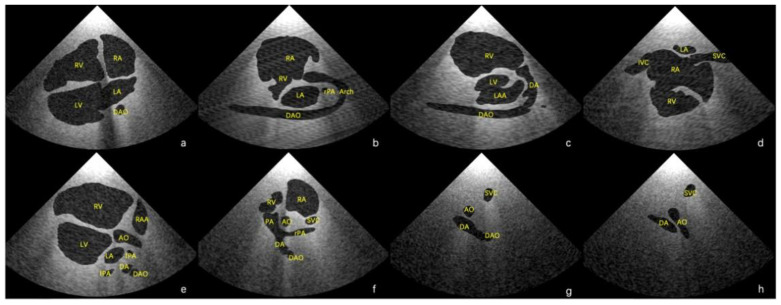
Eight main standard virtual views of echocardiography in Case 1. (**a**) Four-chamber view, (**b**) aortic arch view, (**c**) ductal view, (**d**) bicaval view, (**e**) left ventricular outflow tract view, (**f**) right ventricular outflow tract view, (**g**) three-vessel view, and (**h**) three vessels and trachea view. AO—aorta; Arch—aortic arch; DA—ductus arteriosus; DAO—descending aorta; IVC—inferior vena cava; LAA—left atrial appendage; LV—left ventricle; PA—pulmonary artery; RA—right atrium; RAA—right atrial appendage; RV—right ventricle; SVC—superior vena cava.

**Figure 4 bioengineering-09-00524-f004:**
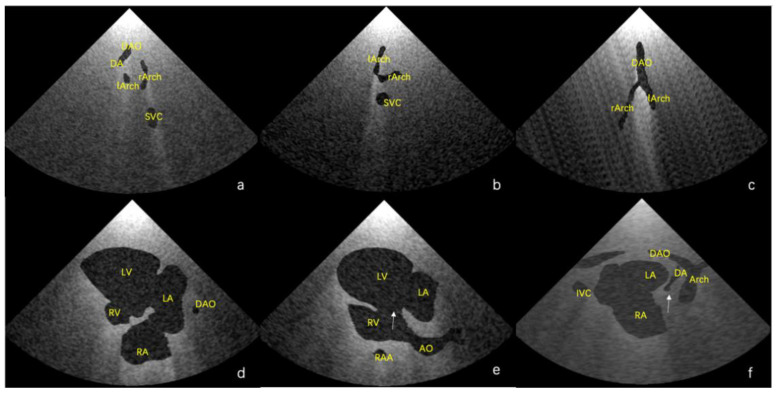
(**a**–**c**) Lesion-sensitive virtual views of the double-aortic arch for Case 2; and (**d**–**f**) hypoplasia of the right heart, ventricle septal defect, malposition of the great arteries, and pulmonary atresia for Case 3.

**Figure 5 bioengineering-09-00524-f005:**
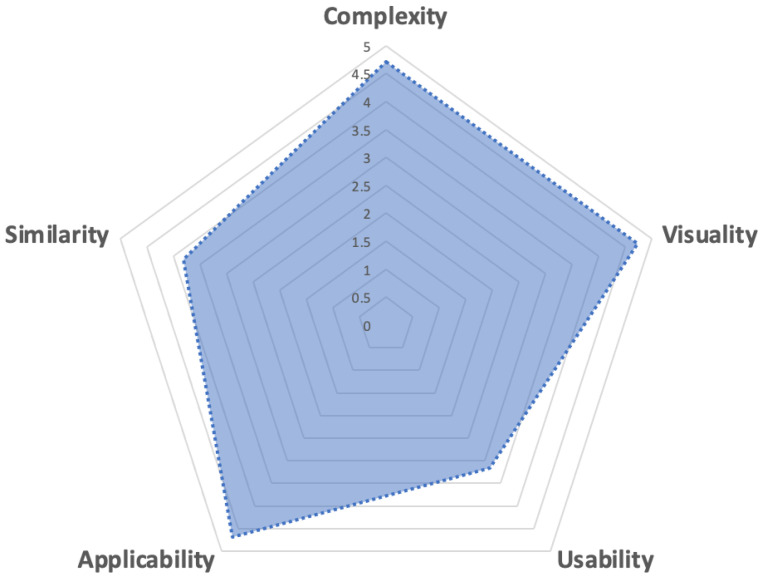
Subjective evaluation results of the feasibility test.

**Table 1 bioengineering-09-00524-t001:** Diagnostic results of the three fetal heart specimens.

Case	GA (Week)	Diagnosis
1	26	Normal heart structure with heart failure
2	26	Double-aortic arch (DArch)
3	25	Hypoplasia of the right heart (HRH) Ventricle septal defect (VSD) Malposition of the great arteries (MGA) Pulmonary atresia (PA)

## Data Availability

The data presented in this study are available upon request from the corresponding author.

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
