# Peer review of "Development of Digital Fetal Heart Models with Virtual Ultrasound Function Based on Cardiovascular Casting and Computed Tomography Scan"

_bioengineering, 2022, doi:10.3390/bioengineering9100524_

Round 1
Reviewer 1 Report
The paper is interesting and well written. Some language revisions are needed.
Good work
Author Response
Response Letter
We would like to thank the reviewer for the professional technical review and constructive comments, many of which are very useful in terms of improving the approaches presented in the current study and useful for directing future research. Moreover, we greatly appreciate the reviewer’s general consent to the value of this research. We have studied the comments carefully and have made corrections which we hope meet with the reviewers’ approval. We provide a response below and include a revised manuscript highlighting all changes that were made.
Response to the comment: Some language revisions are needed.
We thank the reviewer for the comment. Proofreading has been made and a few grammatical mistakes have been corrected.

Reviewer 2 Report
A very interesting, educational and well written manuscript that has clinical importance and merit. However, there are a few editing issues that the authors should consider and address. The following are suggestions/comments regarding these issues. Lines 51 & 52, "...training needs in terms of practice skills. For hands-on ...". Line 54, "...of fetal morphology, which enables training and ...". Lines 93 & 94, "... were exposed to the peeled off thymus. A plastic ...". Line 260, "...similar to that of clinical fetal echocardiography, although ...". Line 262, "...noted that not all clinical fetal ultrasound images ...". Line 267, "...abdominal wall and allowances for the movements of the heart."
Author Response
Response Letter
We would like to thank the reviewer for the professional technical review and constructive comments, many of which are very useful in terms of improving the approaches presented in the current study and useful for directing future research. Moreover, we greatly appreciate the reviewer’s general consent to the value of this research. We have studied the comments carefully and have made corrections which we hope meet with the reviewers’ approval. We provide a response below and include a revised manuscript highlighting all changes that were made.
Response to the comment: However, there are a few editing issues that the authors should consider and address. The following are suggestions/comments regarding these issues. Lines 51 & 52, "...training needs in terms of practice skills. For hands-on ...". Line 54, "...of fetal morphology, which enables training and ...". Lines 93 & 94, "... were exposed to the peeled off thymus. A plastic ...". Line 260, "...similar to that of clinical fetal echocardiography, although ...". Line 262, "...noted that not all clinical fetal ultrasound images ...". Line 267, "...abdominal wall and allowances for the movements of the heart."
These editing issues have been corrected as can be seen in Lines 50-53, 98-99, 264-267,268 and273 in the revised manuscript.
Reviewer 3 Report
It is a very clear and simple document. I should also say a nice article: the goal is the development of an educational tool for learning fetal echocardiography.
the methodology is appropriate and the limits honestly discussed: in particular the absence of cardiac movement, the difference between the size in utero and post mortem of the cavities...
This results in good proposals for the future improvement of this educational tool.
Author Response
Response Letter
We would like to thank the reviewer for the professional technical review and constructive comments. Moreover, we greatly appreciate the reviewer’s general consent to the value of this research. In the revised version, proofreading has been made and a few grammatical mistakes have been corrected.

Reviewer 4 Report
The Authors submitted an interesting Article regarding the development of digital fetal heart models with virtual ultrasound function based on cardiovascular casting and computed tomography scans.
The manuscript is well written and presented; nevertheless, it should be paid more attention to English grammar and structure. Because of clarity, the Authors should specify inclusion/exclusion criteria and how the specimens were selected. In addition, it should be specified how, when, and why the CT scans were conducted. Finally, please specify why the fetus had the exitus.
Author Response
Response Letter
We would like to thank the reviewer for the professional technical review and constructive comments, many of which are very useful in terms of improving the approaches presented in the current study and useful for directing future research. Moreover, we greatly appreciate the reviewer’s general consent to the value of this research. We have studied the comments carefully and have made corrections which we hope meet with the reviewers’ approval. We provide a response below and include a revised manuscript highlighting all changes that were made.
Response to the comment: it should be paid more attention to English grammar and structure。
We thank the reviewer for the comment. Proofreading has been made and a few grammatical and structure mistakes have been corrected.
Response to the comment: Because of clarity, the Authors should specify inclusion/exclusion criteria and how the specimens were selected.
Inclusion criteria: all the fetuses with congenital heart disease (CHD), that the parents finally chose to terminate the pregnancy and to induce labor in our hospital. Moreover, after informed consent, the parents agreed to donate the fetal remains to the center for further research. In our revised manuscript, we have clarified this point with few sentences added to Section 2.1.
Response to the comment: it should be specified how, when, and why the CT scans were conducted.
CT scans were performed in order to produce volumetric data and therefore enable the following steps to generate virtual ultrasound image. To do so, the cast models were scanned by CT, and the 3D volumetric model were generated by manual annotation and segmentation according to the CT image data. The three casts were respectively placed flat in a SOMATOM Definition Flash scanner and scanned with setting the slice thickness as 0.6 mm and distance as 0.4 mm. The CT scan can be obtained at any time after the establishment of the casts. Since the relevant structures have been preserved by the casts, the scan time does not affect the results. In our revised manuscript, we have clarified these issues with few sentences added to Section 2.2.
Response to the comment: please specify why the fetus had the exitus.
The fetus had the exitus because of the child's poor prognosis and/or the parents' desire to have a perfectly normal child, and the parents eventually chose to terminate the pregnancy. In our revised manuscript, we have clarified this point with few sentences added to Section 2.1.

Round 2
Reviewer 4 Report
The manuscript significantly improved after the revision. I have no further comments or edits.